# XPD stalled on cross-linked DNA provides insight into damage verification

Jochen Kuper [1,4] ✉, Tamsanqa Hove[1,4], Sarah Maidl[1], Hermann Neitz [2], Florian Sauer [1], Maximilian Kempf[1], Till Schroeder [1], Elke Greiter[1], Claudia Höbartner [2,3] & Caroline Kisker [1] ✉

The superfamily 2 helicase XPD is a central component of the general transcription factor II H (TFIIH), which is essential for transcription and nucleotide excision DNA repair (NER). Within these two processes, the helicase function of XPD is vital for NER but not for transcription initiation, where XPD acts only as a scaffold for other factors. Using cryo-EM, we deciphered one of the most enigmatic steps in XPD helicase action: the active separation of double-stranded DNA (dsDNA) and its stalling upon approaching a DNA interstrand cross-link, a highly toxic form of DNA damage. The structure shows how dsDNA is separated and reveals a highly unusual involvement of the Arch domain in active dsDNA separation. Combined with mutagenesis and biochemical analyses, we identified distinct functional regions important for helicase activity. Surprisingly, those areas also affect core TFIIH translocase activity, revealing a yet unencountered function of XPD within the TFIIH scaffold. In summary, our data provide a universal basis for NER bubble formation, XPD damage verification and XPG incision.

The preservation of genetic information is constantly challenged by endogenous and exogenous agents. To protect genetic information, efficient countermeasures have evolved[1]. Among these, nucleotide excision repair (NER), a template-based repair mechanism, is exceptional because of its broad substrate specificity. Substrates can range from adducts such as acetylaminofluorene and cisplatin DNA cross-links to cyclobutane pyrimidine dimers (CPDs) and 6–4 photoproducts (6–4 PPs)[2,3]. NER constitutes a multistep multiprotein cascade that can be divided into four distinct phases. The first phase comprises initial lesion recognition and demarks the two possible entry points of the NER cascade[4–6]. At one entry point, transcription-coupled repair (TC-NER), RNA polymerase II (RNA Pol II) becomes stalled upon encountering a lesion. At the other entry point, global genome repair (GG-NER), the XPC complex consisting of XPC, Rad23 and Centrin 2 constantly scans the genome. Once a lesion is encountered in TC-NER or GG-NER, phase two is initiated with the recruitment of general transcription factor II H (TFIIH) unifying the two entry branches. TFIIH is a ten-subunit complex

that can be divided into the core (XPB, XPD, p62, p52, p44, p34 and p8) and the cyclin-dependent kinase (CDK)-activating kinase complex (CAK, consisting of CDK7, MAT1 and Cyclin H); this transcription factor was shown to be essential for transcription and NER[7]. Initially, TFIIH, mainly driven by the XPB translocase, opens a bubble around the lesion. This is aided by the arrival of XPA, enhancing XPB activity and releasing CAK from TFIIH, thereby activating XPD and initiating phase 3, the lesion verification step[6]. Here, the helicase activity of XPD further opens the bubble and, when XPD encounters a lesion, it stalls, enabling the maturation of the preincision complex. Phase 4 is initiated with the 5′ phosphodiester incision by the XPF–ERCC1 nuclease complex mainly positioned by XPA and XPB. Subsequently, gap-filling DNA synthesis powered by proliferating cell nuclear antigen (PCNA), replication factor C (RFC) and DNA polymerase δ is commenced and triggers 3′ incision by the XPG nuclease, which is associated early on with the XPD subunit of TFIIH[5]. Defects in NER can lead to severe diseases such as xeroderma pigmentosum (XP), trichothiodystrophy

[1]Rudolf Virchow Center for Integrative and Translational Bioimaging, University of Würzburg, Würzburg, Germany. [2]Institute of Organic Chemistry, University of Würzburg, Würzburg, Germany. [3]Center for Nanosystems Chemistry (CNC), University of Würzburg, Würzburg, Germany. [4]These authors contributed equally: Jochen Kuper, Tamsanqa Hove. ✉e-mail: jochen.kuper@virchow.uni-wuerzburg.de; caroline.kisker@virchow.uni-wuerzburg.de

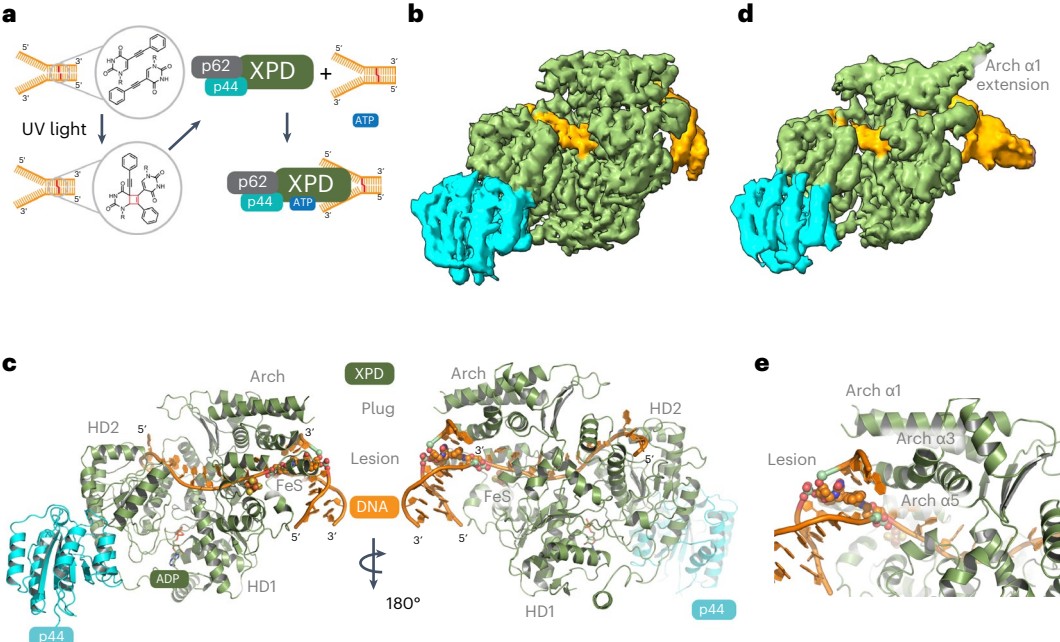

**Fig. 1 | Cryo-EM structure of the XPD complex in the presence of a Y-forked DNA substrate containing an engineered cross-link. a**, Schematic description of sample preparation prior to vitrification. **b**, Cryo-EM map of the class 1 XPD–p44–DNA complex. XPD is colored in green, p44 is colored in cyan and DNA is colored in orange. **c**, Left, structural model of the XPD–p44–DNA complex in cartoon representation, color-coded as in **b**. Right, Rotation (180°) of the model around its *y* axis. **d**, Cryo-EM map of the class 2 XPD–p44–DNA complex, color-coded as in **b**. **e**, Close-up of the dsDNA–ssDNA junction and the cross-link at the Arch domain of XPD, color-coded as in **b**. The cross-link is shown as spheres.

(TTD) and Cockayne syndrome (CS). The hallmark of XP is ultraviolet (UV) light sensitivity with a highly increased incidence of skin cancers, whereas TTD and CS patients suffer from learning disabilities, premature aging and photosensitivity[8].

Recent structural advances on higher-order NER protein complexes have greatly advanced our knowledge of how core TFIIH engages with undamaged Y-forked DNA, as well as how the XPC complex and XPA in GG-NER or RNA Pol II, CSA, CSB and UVSSA in TC-NER synergize to prepare TFIIH for bubble opening and lesion verification[9–12]. For GG-NER, these studies established how core TFIIH is loaded and prepared for bubble opening by the XPC complex and XPA[11,12]. Kokic et al. provided insights into how core TFIIH interacts with Y-forked DNA structures representing the 5′ side of the incision bubble junction[9]. However, vital information on how XPD and, thus, core TFIIH interact with a DNA junction at the 3′ prime side of the bubble and how damage is encountered has not been shown so far.

We determined cryo-EM structures of an XPD–p44–p62 complex from *Chaetomium thermophilum* in the presence of a Y-forked DNA structure containing an engineered interstrand cross-link at 3.1 Å resolution. Combined with functional analyses, our data show how XPD engages with DNA in the unwinding cycle, revealing an unusual double active DNA opening mechanism, placing the Arch domain of XPD as the central player. Furthermore, we identified an unexpected role of the XPD Arch domain for TFIIH translocase activity. Most importantly, our data reveal how XPD approaches DNA damage during the unwinding cycle and how stalling occurs on interstrand cross-linked DNA, leading to a unified model for the excision bubble and highlighting how damage verification followed by incision can be achieved in NER.

## XPD–p44–p62 cross-linked DNA complexes

We heterologously expressed and purified all core TFIIH subunits from *C. thermophilum* as previously described[13]. Core TFIIH proteins from *C. thermophilum* are highly conserved with respect to their human homologs (human XPD and ctXPD share 74% homology and 55% identity) and several other studies validated the *C. thermophilum* proteins

as models for their human counterparts[13–17]. For DNA complex formation, we used equimolar amounts of XPD and the p44–p62 complex, resulting in the heterotrimeric XPD–p44–p62 complex (XPD complex) at 10 μM concentration. As DNA substrate, we used a Y-forked substrate with an interstrand cross-link positioned five bases downstream into the double-stranded DNA (dsDNA) region from the unpaired junction, serving as a noncanonical but bona fide NER substrate[18] (Fig. 1a; see Methods for details). We have shown previously that DNA containing a precursor of this crosslink represents a substrate for the XPD complex that is readily unwound but in its cross-linked form can only be unwound until the cross-link is encountered[19] (Fig. 1a). Protein and DNA were mixed at a molar ratio of 1:1.25 and adenosine triphosphate (ATP) was added to the mixture to initiate unwinding of the substrate. Samples were incubated for 10 min at room temperature and then vitrified for cryo-EM data collection. After data collection and processing (Extended Data Fig. 1), we obtained two volumes (class 1 and class 2) of the XPD–DNA complex at resolutions of 3.1 and 3.4 Å, respectively, according to the GSFSC (gold-standard (GS) Fourier shell correlation (FSC)) (see Methods for details). The three-dimensional (3D) FSC analysis[20] revealed sphericity values of 0.85 (class 1) and 0.78 (class 2), indicating slight anisotropy for class 1 that was more pronounced in class 2 because of preferential orientations (Fig. 1b,d, Extended Data Fig. 2a–d and Table 1). Both maps were, however, readily interpretable. The two classes had in common that XPD and the N-terminal von Willebrand factor type A (vWA) domain of p44 could be easily identified but p62 and the C-terminal zinc finger domains of p44 were unresolved in the density. This was most likely because of the high flexibility of p62 without p34, where the latter serves as an additional anchor within core TFIIH[21] and was also previously observed for p62 with core TFIIH in the presence of DNA[9]. We built the model using class 1. Model and data statistics are provided in Table 1. We can clearly observe the translocating strand in the 5′–3′ direction extending between helicase motor domains 1 and 2 (HD1 and HD2), passing the iron sulfur cluster (FeS) domain and prolonging through the unique XPD pore feature (Fig. 1b–d and Extended Data Fig. 3). After leaving the pore, this strand leads into one

**Table 1 | Cryo-EM data collection, refinement and validation statistics**

| | Class 1 (EMD-19109), (PDB 8REV) | Class 2 (EMD-19109) |
|---|---|---|
| **Data collection and processing** | | |
| Magnification | 105,000 | 105,000 |
| Voltage (kV) | 300 | 300 |
| Electron exposure (e⁻ per Å²) | 49.7 | 49.7 |
| Defocus range (µm) | −1 to −2 | −1 to −2 |
| Pixel size (Å) | 0.84 | 0.84 |
| Symmetry imposed | C1 | C1 |
| Initial particle (no.) | 3,784,237 | 3,784,237 |
| Final particle (no.) | 237,064 | 121,289 |
| Map resolution (Å) | 3.1 | 3.4 |
| FSC threshold | 0.143 | 0.143 |
| Map resolution range (Å) | 3.1–7.2 | 3.4–9.4 |
| **Refinement** | | |
| Model resolution (Å) | 3.4 | |
| FSC threshold | 0.5 | |
| Map sharpening B factor (Å²) | −100 | deepEMhancer |
| Model composition | | |
| Non-hydrogen atoms | 7,797 | |
| Protein residues | 899 | |
| Ligands | 3 | |
| DNA | 24 | |
| B factors (Å²) | | |
| Protein | 1,153 | |
| Ligand | 115 | |
| DNA | 293 | |
| R.m.s.d. | | |
| Bond lengths (Å) | 0.007 | |
| Bond angles (°) | 1.003 | |
| Validation | | |
| MolProbity score | 2.7 | |
| Clashscore | 16.1 | |
| Poor rotamers (%) | 5.7 | |
| Ramachandran plot | | |
| Favored (%) | 93.8 | |
| Allowed (%) | 5.9 | |
| Disallowed (%) | 0.2 | |

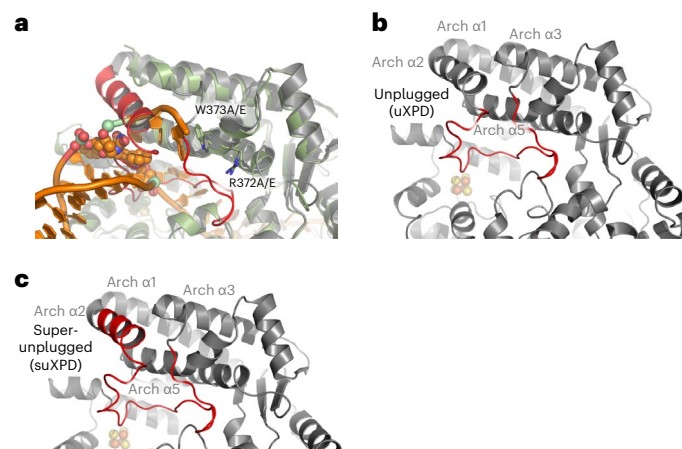

**Fig. 2 | Structure-based functional mutagenesis. a**, Superposition of the XPD Arch domain in complex with cross-linked DNA and apo XPD (PDB 6NMI) in cartoon representation. Apo XPD is colored in gray with the plug region colored in red; XPD from this work is colored in green. Conserved residues subjected to mutagenesis are shown in stick representation. **b**, Apo XPD as in **a**. The loop region that was deleted to create uXPD is shown in red. Four amino acids (S-T-G-S) were inserted to bridge the gap. **c**, As in **b** except that suXPD was generated with the additional removal of α2 and the insertion of three residues (S-G-S) to fill the gap.

and Extended Data Fig. 2f). However, lowering the density thresholds for ADP indicated that an ADP–ATP mixture could also be present. The interstrand cross-link created by the cyclobutene PhedU dimer is reasonably defined in the density. The two phenyl rings, however, could not be visualized in the sharpened density, probably because of lower resolution in that region caused by the higher flexibility of the DNA and Arch domain (Extended Data Fig. 2b,e). Furthermore, the so-called plug region of XPD is disordered[9]. Class 2 mainly reveals similar features but could be resolved to a resolution of only 3.4 Å. The dsDNA region exhibits higher mobility and fewer features in the density. One major difference between the two classes, however, is an extended density that could be attributed to Arch α1, which seems to notably elongate this helix as compared to class 1 and other DNA-containing core TFIIH XPD structures[9,11]. This extension could indicate where the plug feature of XPD moves in the presence of DNA (Fig. 1b,d).

## Functional characterization of Arch domain elements

Our structure suggests that Arch α5 might contribute to unwinding of the dsDNA by protein–DNA interactions. We, therefore, individually substituted the conserved residues W373 and R372 located in Arch α5 to alanine and glutamate resulting in variants W373A/E and R372A/E (Fig. 2a). W373 and R372 could be potentially involved in base and backbone interactions, respectively, although the DNA in our structure can be observed only up to W373. In addition, we investigated the role of the plug element and generated the unplugged XPD variant (uXPD) where we removed the loop region (residues 292 to 315) and the super-unplugged XPD variant (suXPD) where we additionally removed most of Arch α2 (residues 281 to 315) (Fig. 2b,c). All resulting variants were purified to homogeneity and thermal stability analysis confirmed correct folding comparable to wild-type (WT) XPD (Fig. 3a). The highest deviation from the melting point of WT XPD was observed for W373A (−6 °C) and R372A (−7 °C). However, these variants still show highly cooperative unfolding, WT-like DNA binding and ATPase activities, indicating no major structural changes. We subjected all variants to a detailed biochemical analysis, that is, DNA binding, ATPase activity and helicase activity (Fig. 3, Extended Data Table 1 and Extended Data Fig. 4).

turn of dsDNA (Fig. 1c and Extended Data Fig. 3). The complementary strand extends to the single-stranded DNA (ssDNA)–dsDNA junction with three bases that separate into ssDNA at the Arch domain (Fig. 1c,e). Overall, we can observe 18 bases of the translocating strand and 8 bases of the nontranslocating strand including the cross-link (Extended Data Fig. 2e). The dsDNA opening area is located at Arch α3 and α5, which also reveals the location of the DNA cross-link adjacent to theses helices, indicating that the XPD complex unwound the dsDNA until it encountered the cross-link (Fig. 1e). This is further supported by the sequence assignment of the translocating strand and our observation of adenosine diphosphate (ADP) in the ATP-binding pocket, suggesting that ATP hydrolysis and partial unwinding was performed (Fig. 1c

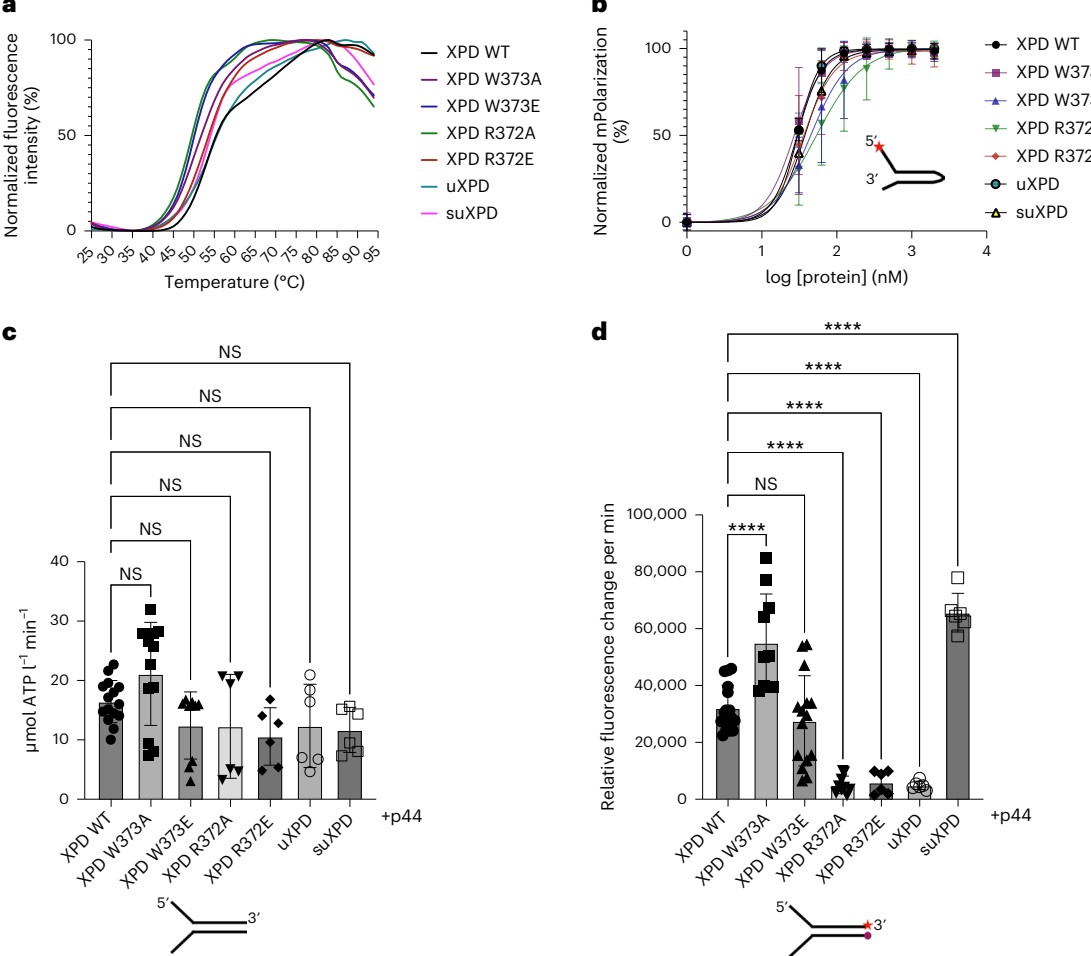

**Fig. 3 | Functional characterization of XPD variants. a**, Normalized thermal unfolding curves of XPD and XPD variants analyzed in this work. Melting points were derived from these curves using GraphPad Prism. **b**, Binding curves obtained from fluorescence anisotropy experiments using a 5′ overhang hairpin substrate. The curves were fitted using GraphPad Prism, resulting in the $K_D$ values given in Extended Data Table 1. Experiments were performed in at least three technical replicates. Mean values are plotted with their associated s.d. The red star marks the Cy3 label. **c**, ATPase activity of XPD and its variants in the presence of a Y-forked substrate and p44. Experiments were performed in at least three technical replicates and one biological replicate. NS, not significant. **d**, Helicase activity of XPD and its variants in the presence of a fluorescently

labeled Y-forked substrate and p44. The red star denotes the Cy3 label at the 3′ end that is quenched by a Dabcyl moiety at the 5′ end of the complementary strand. Experiments were performed in at least three technical replicates and one biological replicate. Data were analyzed using GraphPad Prism. All values are also listed in Extended Data Table 1. Asterisks indicate significance determined by ordinary one-way analysis of variance (ANOVA) testing in GraphPad Prism. ****$P > 0.0001$. All error bars represent the s.d. Number of samples: WT XPD, **b** ($n = 10$), **c** ($n = 15$) and **d** ($n = 16$); XPD W373A, **b** ($n = 10$), **c** ($n = 12$) and **d** ($n = 10$); XPD W373E, **b** ($n = 10$), **c** ($n = 9$) and **d** ($n = 15$); XPD R372A, **b** ($n = 10$), **c** ($n = 6$) and **d** ($n = 9$); XPD R372E, **b** ($n = 10$), **c** ($n = 6$) and **d** ($n = 6$); uXPD, **b** ($n = 3$), **c** ($n = 6$) and **d** ($n = 6$); suXPD, **b** ($n = 3$), **c** ($n = 6$) and **d** ($n = 6$).

A 5′ overhang hairpin substrate was used for the interaction with DNA, which bound with high affinity to WT XPD ($K_D = 30$ nM). The $K_D$ values of the variants ranged from 27 to 54 nM, indicating no substantial influence on DNA binding (Fig. 3b and Extended Data Table 1). WT XPD displayed a robust ATPase activity of 16 μM ATP min⁻¹ in the presence of Y-forked DNA and p44. For all variants tested, the ATPase activity did not deviate substantially from WT, indicating no influence on ATPase activity of each variant (Fig. 3c and Extended Data Table 1). Overall, our data indicate that DNA binding and ATPase activity were not or only slightly affected in all investigated variants. Substantial differences between the WT protein and the variants, however, were observed with respect to their helicase activity using a Y-forked substrate (Fig. 3d and Extended Data Fig. 4). W373A showed enhanced activity that amounted to 172%, whereas the glutamate variant, W372E, displayed only a small decrease to 88% of WT activity. Both R372 variants showed a strong decrease in helicase activity (16% and 19%), indicating that this residue is highly relevant for dsDNA separation. uXPD also displayed a substantial decrease in helicase activity (16%), whereas the additional removal of

Arch α2 in suXPD led to the highest helicase activity with 206% compared to the WT protein (Fig. 3d). The importance of Arch α5 is further strengthened by the presence of the XPD TTD variant R378H (located in Arch α5)[22]. Interestingly, missense mutations encoding residues in Arch α5 can be found in the cBioportal database (www.cBioportal.org) from studies of different cancer entities, indicating that there could be a functional role associated. Overall, the R378H variant and the cancer-associated cBioportal data support the notion that Arch α5 is an important element for XPD function.

## The unusual role of the Arch domain for XPD helicase action

Our data clearly indicate that the Arch domain is essential for XPD helicase action (Fig. 3). The total removal of the plug region (suXPD) led to a hyperactive helicase, whereas removing only the loop region impaired helicase activity (uXPD). In our DNA-bound structure, the plug region is disordered; thus, we could not observe any interaction of the plug with the dsDNA. Our class 2 data, however, clearly show an

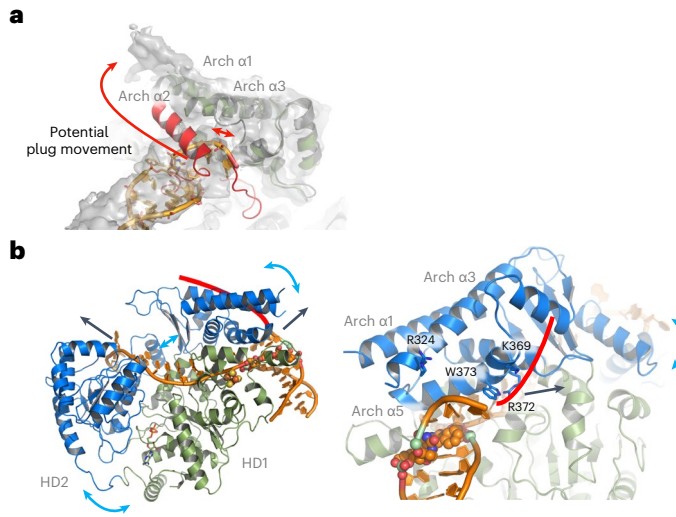

**Fig. 4 | XPD helicase mechanism and plug dynamics. a**, Suggested movement of Arch α2 to elongate Arch α1, explaining the additional density observed in our class 2 data. The long red arrow indicates the motion of the plug, whereas the short red double arrow shows why the movement could be hindered in uXPD. **b**, Two views of XPD in complex with the cross-linked DNA substrate in cartoon representation. HD2 and the Arch domain are colored in blue and the remainder of XPD is colored in green. DNA is shown in orange. Blue arrows indicate the possible domain movement during ATP hydrolysis and ssDNA translocation. Black arrows indicate the direction of the DNA movement. The red line provides the likely path of the nontranslocated strand along the Arch domain supported by functional data. Left, overview focusing on the ssDNA path. Right, view focusing on the XPD pore entry where the dsDNA is separated.

elongation of Arch α1 that may be attributed to Arch α2 moving up and 'fusing' with Arch α1 when DNA is bound (Fig. 4a). To accommodate this movement, the entire loop region of the plug has to undergo a conformational change to move upward with the helix. Removal of the loop in uXPD restricts this movement, trapping Arch α2 in position and hindering helicase activity; this strengthens the notion of the plug acting as a negative regulatory element[9]. Interestingly, Arch α2 and the plug shield Arch α5, in which we previously located functionally important residues for helicase action. Activating XPD, therefore, requires an upward and outward movement of the plug, potentially leading to interactions with other proteins required in the NER pathway such as XPG[9] (Fig. 4a).

With the plug in an outward position, XPD is now primed for helicase activity. It was established for the bacterial XPD homolog DinG that ATP-dependent ssDNA translocation mediated by HD1 and HD2 movement is translated into a swingout motion of the Arch domain by a pusher helix in HD2 (ref. 23) (Fig. 4b). According to our structure, the proposed swingout of the Arch domain would lead to a highly unusual active pulling on the nontranslocated strand that is separated from the dsDNA at the Arch domain DNA junction (Fig. 4b). This mechanism would actively engage both the translocating and the nontranslocating strand to promote the separation of dsDNA. This hypothesis is further supported by the presence of essential residues that modulate helicase activity (Fig. 4b, lower panel)[14]. Our analysis showed that R372 is vital for activity, possibly because of phosphate backbone interactions, whereas W373 may be involved in base interactions, which, if removed, could increase DNA separation capacity. In a previous study, we identified R324 and K369 to be essential for helicase activity[14]. R324 is located in close vicinity to the phosphate backbone of the nontranslocating ssDNA strand and K369 lies in the proposed path for the nontranslocating strand across the Arch domain (Fig. 4). Importantly, this proposed mechanism is most likely conserved in all XPD homologs including

human RTEL1, FANCJ and DDX11, which all contain Arch domains that could interact with DNA in a similar fashion (Extended Data Fig. 5)

## An unexpected role for XPD in the TFIIH translocase complex

A superposition of our XPD–p44 DNA structure with the core TFIIH structure revealed no notable differences in the ssDNA interactions with respect to the translocated strand (Fig. 5a). Furthermore, the overall orientation of XPD–p44 is highly comparable to that observed in core TFIIH (root-mean-square deviation (r.m.s.d.) of 1.7 Å), indicating that our structural and functional analysis can be readily transferred to core TFIIH.

All prior data suggested that XPD is not involved in the dsDNA translocase activity of core TFIIH[13,24]. To our surprise, however, we observed that uXPD and suXPD modulate core TFIIH dsDNA translocase activity. suXPD displays a substantially increased activity of 286% compared to WT XPD, whereas uXPD shows a slightly decreased activity (62% of WT XPD), indicating a strong influence of the plug region on core TFIIH translocase when removed completely (Fig. 5b). Notably, it is unlikely that XPD engages with ssDNA in core TFIIH dsDNA translocase activity. Thus, our results demonstrate an unexpected dsDNA interaction of XPD that boosts core TFIIH translocase. This observed boost upon plug removal is likely to have potential implications for NER but not for the transcriptional TFIIH translocase where holo TFIIH is involved and XPD helicase is inhibited by the CAK complex[15,24–26]. For the repair process, it was suggested that TFIIH could be involved in pushing back RNA Pol II before damage verification in TC-NER[27,28]. A highly active translocase might be necessary at this stage to facilitate the pushback and engage core TFIIH with the damage location. This is supported by the observation that repair factors such as XPA and XPG also increase core TFIIH translocase and helicase activity[9,29]. XPA could facilitate CAK removal, thus enhancing plug flexibility and permitting the interaction of the Arch domain with XPG[14]. This interaction could result in an open plug conformation, as indicated in the class 2 data (Fig. 4b), which is essentially mimicked by suXPD.

To investigate the potential interaction between XPD and XPG, we pursued XPD helicase activity analyses in the presence of equimolar amounts of incision-incompetent XPG D924A. The addition of XPG D924A led to XPD helicase activation (Fig. 5c and Extended Data Table 2), in line with previous data[9,30]. In the presence of uXPD, this behavior was similar and we observed substantial activation upon XPG D924A addition. However, the 'hyperactive' suXPD variant was strongly inhibited (sixfold) by the addition of XPG D924A (Fig. 5c), leading to the conclusion that there is cross-talk between the plug region of XPD and XPG, which regulates XPD activity. These observations suggest that the plug region could be involved in XPD–XPG signaling with respect to when the incision can be made by XPG (as previously observed[30]), adding an additional layer of complexity to the Arch domain and plug function.

## Implications for NER damage verification

Our data provide insights into the formation of the NER bubble prior to the two incisions. A merge of our structure with the core TFIIH–XPA–DNA structure[9] (Protein Data Bank (PDB) 6RO4) led to a stretch of 11 bases of ssDNA spanning the XPD helicase, which is framed on both sides by the two dsDNA junctions of the NER bubble (Fig. 5d). The proposed route of the nontranslocating strand bridges the two ssDNA ends of both structures where the DNA could not be resolved because of high flexibility. This model could likely represent an early stage of bubble opening by the XPB–XPA translocase activity[11], directly followed by XPD engagement and initial unwinding. This initial bubble then requires widening before the primary 5′ incision because both endonucleases incise at a ssDNA–dsDNA junction[4]. The excised fragment spans on average ~27 bases with the damage being ~5 bases away from the 3′ end[31]. Widening is likely facilitated by XPD, which separates the dsDNA until it is stalled by a lesion. In our structure, the lesion is

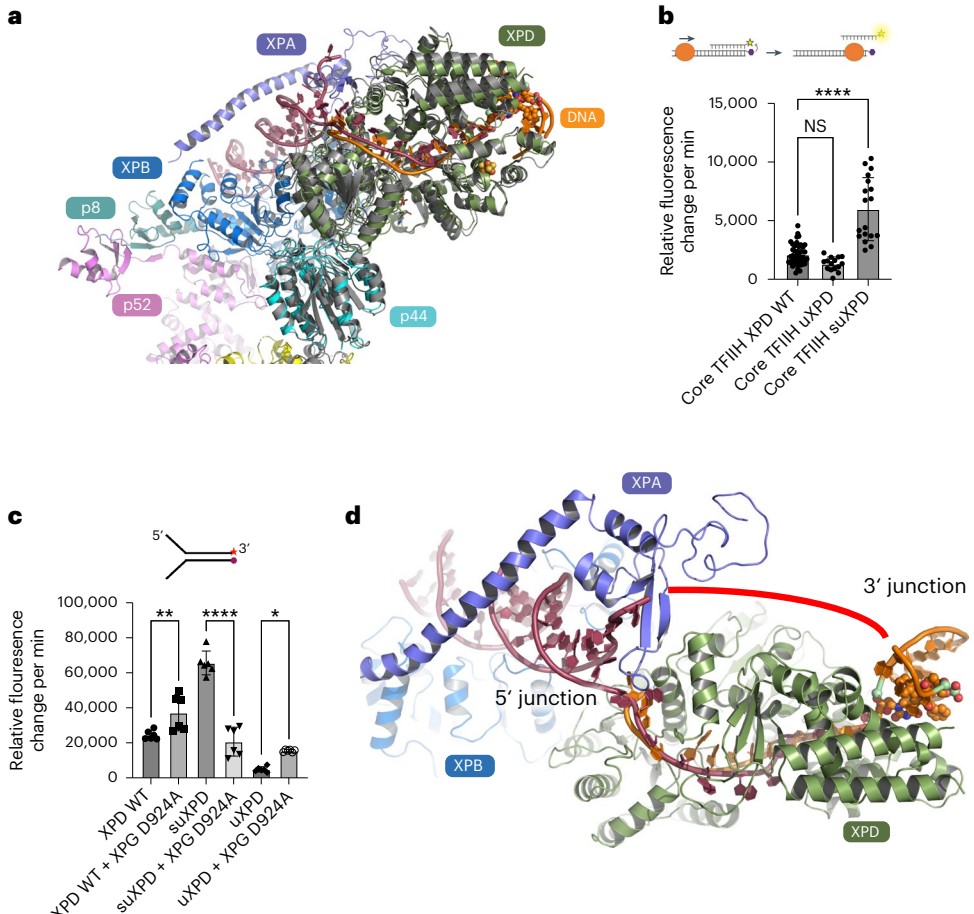

**Fig. 5 | XPD cross-linked structure integrates into core TFIIH. a**, Superposition of the XPD–p44–DNA complex (color-coded as in Fig. 1c) from this work with XPD–p44 (colored in gray) in core TFIIH bound to DNA (PDB 6RO4). The DNA from PDB 6RO4 is colored in dark red. **b**, Translocase activity of XPD and plug variants in a triplex disruption assay. Experiments were performed in at least three technical replicates and one biological replicate. Data were analyzed using GraphPad Prism. All values are also provided in Extended Data Table 1. **c**, Helicase activity of XPD and variants in the absence and presence of XPG D924A. Experiments were performed in at least three technical replicates and one biological replicate. Data for uXPD and suXPD were also used in Fig. 3d. Data were analyzed using GraphPad Prism. All values are also provided in Extended

Data Table 2. **d**, Model of the early incision bubble of core TFIIH, color-coded as in **a**. The XPD structure and orange-colored DNA are from this work, while the remaining structural elements were taken from PDB 6RO4. Both sets of DNA (PDB 6RO4 in firebrick red and our data in orange) combined could form the early bubble. The unresolved part of the DNA is indicated by the red line. Asterisks indicate significance determined by ordinary one-way ANOVA testing in GraphPad Prism. *$P > 0.05$, **$P > 0.005$ and ****$P > 0.0001$. All error bars represent the s.d. Number of samples: **b**, core WT XPD ($n = 47$), core TFIIH uXPD ($n = 15$) and core TFIIH uXPD ($n = 18$); **c**, WT XPD ($n = 6$), WT XPD + XPG D924A ($n = 6$), suXPD ($n = 6$), suXPD + XPG D924A ($n = 6$), uXPD ($n = 6$) and uXPD + XPG D924A ($n = 6$).

an interstrand cross-link, that is, a noncanonical NER substrate. The overall architecture of the lesion site indicates that Arch α5 is not only directly involved in strand separation but also located in close vicinity to the lesion, potentially acting as a sensor (Figs. 4b and 6a). This hypothesis is further supported by recent molecular modeling studies of CPD or 6–4 PP lesions in complex with XPD by Fu et al.[32,33]. Their analysis suggested that helix α5 of the Arch domain acts as a lesion sensor for canonical NER lesions, well in line with the role we observed in our structure (Fig. 6). In addition, we observed interactions of the known lesion-sensing residues Y192 and R196 (Y191 and R195 in ctXPD, respectively)[34] located in the FeS domain with a backbone phosphate that is located adjacent to the first phosphate of the lesion (P-1, Fig. 6a). This is in contrast to the 6–4 PP model where these two amino acids engage phosphates belonging directly to the damaged bases (P0 and P1, Fig. 6b) indicating that the lesion moves toward the XPD pore. With the encounter of the 6–4 PP lesion, the Arch domain locks onto the damage with a rotational movement, completely engulfing the lesion in the pore. Interestingly, the CPD lesion is stalled outside of the pore and the Arch domain engages almost identically to our structure (Fig. 6c).

Single-molecule studies, however, suggested that the XPD Arch domain is also locked in a closed conformation upon CPD stalling[35].

Following from these results, our structural data for the recognition of the noncanonical cross-link lesion may form the basis to rationalize damage recognition of canonical lesions in the context of dsDNA unwinding and helicase stalling. After lesion verification is achieved by XPD, NER proceeds with the 5′ and 3′ incision of the lesion. The XPA–XPB proteins demark the 5′ junction of the NER bubble and XPD–XPG demark the 3′ site. The position of the dsDNA of the substrate enables us to model a possible engagement of XPG to perform the 3′ incision. We superimposed the dsDNA of our structure with the DNA-bound structure of the yeast XPG homolog Rad2 (ref. 36) (PDB 4Q0W), leading to a complex with the catalytic core of Rad2 being in close vicinity to the DNA junction, thus revealing a potential 3′ incision site (Fig. 7). This model is well in line with our biochemical data, showing that the XPD plug interacts with XPG, and with cross-linking data from Kokic et al.[9], where peptides of the plug region were identified to interact with residues of XPG located adjacent to the C terminus of the catalytic region. However, in our model, the complex trapped on the

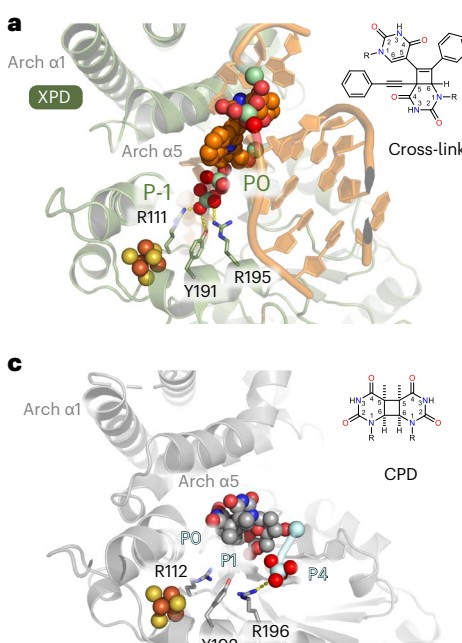

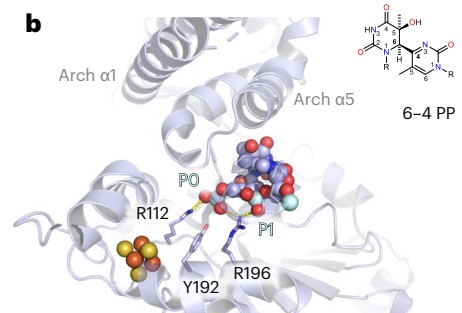

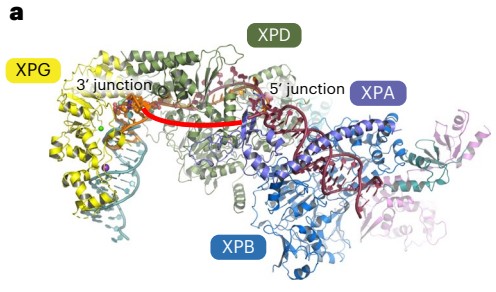

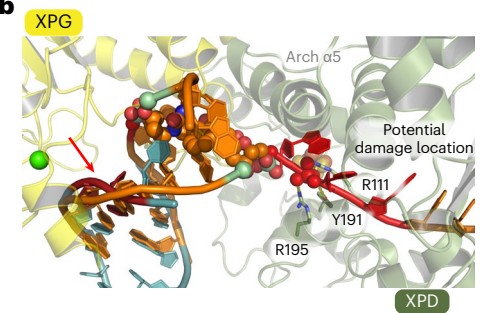

**Fig. 6 | Lesion verification strategies of XPD. a**, Lesion-stalled XPD (green) upon cross-link encounter. XPD and DNA are shown in cartoon mode and the interstrand cross-link lesion is shown as spheres. Relevant backbone phosphate positions relative to the lesion are indicated (P-1 and P0), where P0 marks the first phosphate from the lesion in the 3′ direction. R111, Y191 and R195 from *C. thermophilum* correspond to R112, Y192 and R196 in human XPD. **b**, Lesion-stalled XPD (light blue) upon 6–4 PP lesion encounter. XPD and DNA are shown in cartoon mode and the 6–4 PP lesion is shown as spheres. Relevant backbone phosphate positions relative to the lesion are indicated (P0 and P1). **c**, Lesion-stalled XPD (gray) upon CPD lesion encounter. XPD and DNA are shown in cartoon mode and the CPD lesion is shown as spheres. Relevant backbone phosphate positions relative to the lesion are indicated (P0, P1 and P4). **b** and **c** show representative end states of the molecular dynamics simulations from a previous study[31].

**Fig. 7 | Model of the NER incision bubble. a**, Model of the incision bubble of core TFIIH combined with XPG at the 3′ junction based on the superposition of the XPG (yellow) substrate DNA (cyan) complex with the dsDNA of the XPD–p44–DNA complex modeled in core TFIIH, color-coded as in Fig. 5d. **b**, Close-up of **a**, indicating a possible incision site (red arrow) and location of the canonical damage (dark-red DNA backbone). Note that the cross-link forces the DNA to be closed at the potential incision site. Non-interstrand cross-linked damaged DNA could already be separated at that position.

interstrand cross-link would be stalled nonproductively because XPG does not encounter the ssDNA–dsDNA junction structure required for incision. Interestingly, earlier data showed that archaeal XPD can perform backtracking[37] and this was also more recently observed for human XPD[38]. It is, thus, plausible that XPD, after encountering the lesion and initial stalling, may backtrack because it is not locked onto the lesion as one would expect with respect to the encounter of canonical damages. Backtracking would then enable XPG to incise 5′ to the interstrand cross-link (Extended Data Fig. 6). In fact, XPD-mediated lesion-independent cutting by XPG was recently observed in vitro[30]. The 5′ junction would still be demarked by the XPA–XPB complex. This would lead to a shorter, lesion-free double 5′ incision product, which is in full agreement with earlier studies where this incision pattern was observed[39,40].

In contrast, when a canonical lesion is encountered, XPD is stalled differently. For 6–4 PPs, the Arch domain locks onto the lesion with a rotational movement engulfing the lesion in the pore (Fig. 6b), likely being responsible for further 3′ DNA opening in a dsDNA context. The CPD lesion is stalled outside the pore and the Arch domain engages almost identically to our structure[32,33] (Fig. 6a,c). The main difference to the cross-link is that both canonical lesions do not block the DNA from being further opened by Arch domain dynamics. Thus, additional base separation in the 3′ direction is possible, creating a structure that can be cleaved by XPG (Extended Data Fig. 6). This additional base separation is likely enhanced by XPG itself. However, because the resynthesis machinery induces the 5′ incision[5], cleavage can be triggered by the physical strain that is imposed upon resynthesis on the undamaged template strand, forcing further

separation of the lesion-containing strand that is mainly bound to XPD.

The distance from the proposed 3′ incision site to the position of the lesion in our model is in exact agreement with experimental data showing that the damage is located asymmetrically on the incised fragment with ~5 bases toward the 5′ side, which would match the P0 location for canonical damage recognition (Fig. 7b).

Combined, our data provide vital information on the essential XPD helicase, as well as the other FeS-containing helicases FANCJ, RTEL1 and DDX11, revealing an unusual double active unwinding mechanism engaging both strands. Furthermore, we identified the XPD plug as a regulatory element that not only is involved in helicase function but also actively modulates core TFIIH translocase activity. Lastly, we provided a model of the early NER incision bubble, explaining how canonical and noncanonical lesions are verified and how XPG could be positioned for productive incision.

## Online content

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

## Methods

### Protein expression, purification and mutagenesis

The genes encoding ctXPB, ctXPD, ctp62, ctp52, ctp44, ctp34 and ctp8 were cloned from *C. thermophilum* complementary DNA (cDNA). The cDNA sequence of ctXPB was codon-optimized for expression in *Escherichia coli* (ATG:biosynthetics). CtXPB and ctXPG were inserted into the pFastBac vector (Thermo Fisher) containing a C-terminal Twin-Strep tag and a 10xHis tag. CtXPD, ctp44, ctp34 and ctp8 were inserted individually into the pBADM-11 vector containing an N-terminal 6xHis tag with a tobacco etch virus (TEV) cleavage site. Ctp62 and ctp52 were each inserted into the pETM-11 vector (EMBL) without a tag. CtXPB and ctXPG D924A from the pFastBac vector were expressed using the baculovirus system. Bacmids were prepared in SF21 cells in EX-CELL 420 medium (Sigma-Aldrich) at 27 °C. The cell culture medium containing the viruses was harvested after approximately 72 h. For ctXPB, Hi5 cells were grown in EX-CELL 405 medium (Sigma-Aldrich) at 27 °C until a density of $0.5 \times 10^6$ cells per ml was reached, followed by transfection with 10% (v/v) virus-containing medium. Protein expression was allowed for 72 h at 27 °C. CtXPG D924A was expressed under similar conditions using SF9 cells. CtXPD was expressed in ArcticExpress (DE3) RIL cells (Agilent). Ctp62/ctp44 and ctp52/ctp34 were coexpressed in BL21 CodonPlus (DE3) RIL cells (Agilent). The *E. coli* cells were grown at 37 °C in either Lennox broth (ctp44, ctp8, ctp52–ctp8 and ctp52–ctp34) or Terrific broth (ctXPD and ctp62/ctp44) medium (Carl Roth) to reach a final optical density at 600 nm (OD600) of 0.6 or 1.2, respectively. When the final $OD_{600}$ was reached, protein expression was induced by addition of 0.5 mM IPTG for pETM-11 vectors or 3.3 mM arabinose for pBADM-11 vectors, accompanied by a temperature reduction to 15 °C, and protein expression was allowed overnight. For expression of ctXPD, the temperature was reduced to 30 °C once an $OD_{600}$ of 0.6 was reached and the cells were allowed to grow until an $OD_{600}$ of 1.2 was reached, followed by induction and protein expression at 11 °C.

Ctp44 and ctp62 were copurified by immobilized metal affinity chromatography (IMAC) using Ni-IDA beads (Macherey-Nagel), followed by size-exclusion chromatography (SEC) using a HiLoad 16/600 Superdex 200 prep grade column (Cytiva) with 20 mM HEPES pH 7.5, 250 mM NaCl and 1 mM TCEP. The elution fractions containing ctp44 and ctp62 were pooled and used for anion-exchange chromatography (AEC). For AEC, the mixture was applied to a MonoQ 5/50 GL column (Cytiva), with buffers containing 20 mM HEPES pH 7.5, 50 or 1,000 mM NaCl and 1 mM TCEP.

To obtain ctXPD, the protein was purified by IMAC, followed by SEC and AEC. The SEC buffer contained 20 mM HEPES pH 7.5, 150 mM NaCl, 5 mM MgCl$_2$ and 1 mM TCEP. The AEC buffers contained 20 mM HEPES pH 7.5, 80 or 1,000 mM NaCl, 5 mM MgCl$_2$ and 1 mM TCEP.

CtXPB (Hi5 cells) was purified using a 5-ml StrepTrap HP column (Cytiva) followed by SEC using a HiLoad 16/600 Superdex 200 prep grade column. For SEC, the buffer contained 20 mM HEPES pH 8.0, 200 mM NaCl and 1 mM TCEP.

Ctp52/ctp34, ctp44 and ctp8 were purified by IMAC (Ni TED or Ni IDA; Macherey-Nagel) and SEC using a HiLoad 16/600 Superdex 200 prep grade column (Cytiva). For ctp52/ctp34 and ctp44, the SEC buffer contained 20 mM HEPES pH 7.5, 250 mM NaCl and 1 mM TCEP. For ctp8, the SEC buffer contained 20 mM HEPES pH 8 and 375 mM NaCl.

CtXPG D924A was purified by affinity chromatography using an N-terminal histidine tag and Ni IDA resin (Macherey-Nagel). After IMAC purification, the protein was subjected to SEC on a HiLoad 16/600 Superdex 200 prep grade column (Cytiva) with 20 mM HEPES pH 7.5, 250 mM NaCl and 1 mM TCEP. After SEC, ctXPG D924A was subjected to AEC using a MonoQ 5/50 GL column (Cytiva), with buffers containing 20 mM HEPES pH 7.5, 50 or 1,000 mM NaCl and 1 mM TCEP. All proteins were concentrated to 50–1,000 µM, flash-frozen in liquid nitrogen and stored at −80 °C.

For the generation of the *C. thermophilum* XPD–p44–p62 complex, we mixed equimolar amounts of p44–p62 and XPD. Core TFIIH was

generated by mixing equimolar amounts of the purified components, resulting in active complexes as described before[13]. XPD single-amino acid variants were generated by site-directed mutagenesis[41]. uXPD and suXPD were generated by deleting the sequences encoding residues 292–315 for uXPD (replaced with the linker sequence S-T-G-S) and 281–315 for suXPD (replaced with the linker sequence S-G-S) using sequence-independent and ligation-independent cloning (SLIC)[42]. The N-terminal domain of p44 (residues 1–287) was generated by deleting the C-terminal residues from 288 to the end using SLIC. All variants were purified following the protocol used for the WT proteins without modifications.

### Cryo-EM sample preparation and data collection

$^{Phe}$dU containing DNA to generate the cyclobutene $^{Phe}$dU dimer cross-link by alkene–alkyne [2 + 2] photocycloaddition was produced and annealed as previously described (fork 1, 5′-AGCTACCATGCCTGCACGAATTAAGCA($^{Phe}$dU)CGCGTAATC ATGGTCATAG-3′; fork 2, 5′-CTATGACCATGATTACGC($^{Phe}$dU)CTGCT TGGAATCCTGACGAACTGTAGA-3′)[19]. We mixed the cross-link containing DNA substrate with 10 µM XPD–p44–p62 complex at a molar ratio of 1.25:1. Samples were mixed in 20 mM HEPES pH 7.5, 50 mM KCl, 1 mM TCEP and 5 mM MgCl$_2$, the same buffer used for XPD activity analysis, to ensure efficient partial substrate unwinding (see our previous study[19]). Samples were incubated on ice for 10 min and 5 mM ATP was added to initiate the helicase cycle. The mixture was further incubated for 10 min at room temperature to allow efficient substrate engagement. Samples were subsequently immediately used for cryo-grid preparation. Then 3 µl of sample were applied to glow-discharged R2/2 carbon grids (Quantifoil), which were blotted for 5 s at a blot force of 25 and plunge-frozen in liquid ethane with a Vitrobot Mark IV (Thermo Fisher) operated at 4 °C and 100% humidity. Data were collected at the CM01 facility of the European Synchrotron Radiation Facility (ESRF)[43]. Micrographs were acquired at a nominal magnification of ×105,000 (0.84 Å per pixel) using a dose rate of 17.5 e⁻ per pixel per s over a time of 2 s, resulting in a total dose of 49.7 e⁻ per Å$^2$ fractionated over 50 frames. Movies were recorded over a defocus spread of −1 µm to −2 µm with a 0.2-µm step size. Overall, a total of 24,603 movies were collected.

### Cryo-EM processing and model building

Motion correction and dose weighting were performed using MotionCor2 (ref. [44]) within the CryoSPARC framework[45]. CTF correction was achieved using patch CTF from CryoSPARC[45]. We used the template-based picking algorithm in CryoSPARC with a low-resolution model of the complex that was obtained in house using a Thermo Fisher Titan-Krios G3 with an X-FEG source (300 kV) and a Falcon III camera. Initial picking and two-dimensional classification (fourfold binning) in CryoSPARC resulted in 3,784,237 particles that were subjected to CryoSPARC ab initio modeling with subsequent heterologous refinement of the resulting three classes, which revealed one class containing XPD–p44 with 2,016,863 particles that were subsequently re-extracted with twofold binning. This set was subjected to nonuniform 3D refinement in CryoSPARC and subsequently analyzed using the 3D variability function. The 3D variability analysis resulted in five clusters that were subjected to further heterologous refinement. Of the five classes, two contained dsDNA located at the Arch domain; these classes were pooled and re-extracted at full size (box size: 384 pixels). Nonuniform 3D refinement was performed on this set followed by an additional round of 3D variability analysis and heterologous refinement. In this round, the final class 1 and class 2 data were obtained with 237,064 and 121,289 particles, respectively (see Extended Data Fig. 1 for details and overview). This resulted in an overall resolution of 3.1 Å for class 1 and 3.4 Å for class 2, as defined by the GSFSC 0.143 criterion (Extended Data Fig. 2a,c). Local resolution maps showed the highest resolution for the XPD and ssDNA parts of the density, degrading in the Arch domain and dsDNA regions (Extended Data Fig. 2b,d). For model building, we used

AlphaFold2 models of p44 and XPD from *C. thermophilum* that were assembled on the basis of the arrangement of XPD–p44 in PDB 6RO4. This complex was used for map docking in Phenix[46]. The results of the initial map docking were further improved by manual model building in Coot[47]. No further density was observed that could be attributed to p62 or the C-terminal part of p44. The DNA and the cross-link were built manually in Coot and sequence assignment was based on cross-link location and map quality. Manual model building was iterated with rounds of real-space refinement using the Refmac5 (ref. 48) pipeline in CCPEM[49] and phenix.real_space_refine[46]. The final model and density correlation statistics are given in Table 1.

### In vitro DNA-dependent ATPase activity assay

XPD ATPase activity was measured using an in vitro ATPase assay in which ATP consumption is coupled to the oxidation of nicotinamide adenine dinucleotide (NADH) by pyruvate kinase and lactate dehydrogenase activities, as described previously[14]. The assay was carried out under saturating concentrations of ATP (5 mM) using WT XPD, its variants and p44 (1–287) at a concentration of 250 nM in 20 mM HEPES pH 7.5, 50 mM KCl, 5 mM $MgCl_2$ and 1 mM TCEP. Y-forked DNA (strand 1, 5′-AGCTACCATGCCTGCACGAATTAAGCAATTCGTAATCATGGTCAT AGC-3′; strand 2, 5′-GCTATGACCATGATTACGAATTGCTTGGAATCCT GACGAACTGTAG-3′) was added at a final concentration of 250 nM. The mix of all reagents, with the exception of ATP, was preincubated at 30 °C until a stable baseline was achieved. Enzyme catalysis was initiated by the addition of ATP. The activity profiles were measured at 340 nm using a CLARIOstar (BMG LABTECH) plate reader. Reactions were followed until total NADH consumption was reached. Initial velocities were recorded and ATP consumption was determined using the molar extinction coefficient of NADH. Measurements were carried out with at least three technical replicates and one biological replicate. Mean values were plotted with their associated s.d. The mean and s.d. were determined using GraphPad Prism software.

### In vitro helicase assay

Helicase activity was analyzed using a fluorescence-based helicase assay described previously[14]. We used a Y-forked substrate with a Cy3 label at the 3′ end of the translocated strand (5′-AGCTACCATGCCTGC ACGAATTAAGCAATTCGTAATCATGGTCATAGC-3′-Cy3) and a Dabcyl modification on the 5′ end of the opposite strand (Dabcyl-5′-GCTATG ACCATGATTACGAATTGCTTGGAATCCTGACGAACTGTAG-3′). Assays were carried out in 20 mM HEPES pH 7.5, 50 mM KCl, 5 mM $MgCl_2$ and 1 mM TCEP. DNA concentrations were varied from 500 to 31.25 nM (1:1 dilutions) and proteins were used at a concentration of 250 nM. For Fig. 3d and Extended Data Table 1, only the 250 nM DNA concentrations of these experiments were used. The complete series can be found in Extended Data Fig. 4. For the XPG measurements, XPG D924N was used at 250 nM final concentration with 250 nM DNA substrate and 250 nM XPD–p44. The mix of all reagents, with the exception of ATP, was preincubated at 30 °C until a stable baseline was achieved. The reaction was subsequently started with the addition of 5 mM ATP. Live kinetic measurements were recorded with a CLARIOstar plate reader (BMG LABTECH). Initial velocities of the kinetic data curves were fitted with the MARS software package (BMG LABTECH) and represent the averages of at least three technical replicates and one biological replicate. Mean values were plotted with their associated s.d. The mean and s.d. were determined using GraphPad Prism software.

### Fluorescence anisotropy

DNA binding was analyzed by fluorescence anisotropy using a self-annealing hairpin with a 5′ Cy3 label (Cy3-5′-TTTTTTTTTTTTTTTCCC GGCCATGCGAAGCATGGCCGTT-3′). Assays were carried out in 20 mM HEPES pH 7.5, 50 mM KCl, 5 mM $MgCl_2$, 1 mM TCEP and 5 nM DNA at room temperature. WT XPD and variants were used at concentrations of 31.5–2,000 nM. After mixing, the reaction was incubated for

5 min before recording. Fluorescence was detected at an excitation wavelength of 540 nm and an emission wavelength of 590 nm with a CLARIOstar plate reader (BMG LABTECH). The gain was adjusted to a well containing buffer and DNA but no protein. Curves were fitted with GraphPad Prism and represent the averages of at least three technical replicates. Mean values were plotted with their associated s.d. (Extended Data Fig. 3).

### In vitro translocase activity assay

Core TFIIH dsDNA translocase activity was detected using a well-established triplex disruption assay[9,24]. The dsDNA translocase activity was measured by displacement of a fluorescently labeled triplex-forming oligonucleotide (TFO) from a triple-helix DNA substrate and was carried out as described previously[13] using 150 nM triplex DNA with a black hole quencher (BHQ) used for Cy3 quenching (forward, 5′-GTCTTCTTTTAAACACTATCTTCCTGCT CATTTCTTTCTTCTTTCTTTTCTT-3′; reverse, 5′-BHQ-AAGAAAAGAAA GAAGAAAGAAATGAGCAGGAAGATAGTGTTTAAAAGAAGAC-3′ and 5′-Cy3-TTCTTTTCTTTCTTCTTTCTTTT-3′). Correct triplex formation was confirmed by native polyacrylamide gel electrophoresis (PAGE). The baseline was recorded for 10–15 min before the addition of 2 mM ATP. TFO displacement was measured for 60 min at an excitation wavelength of 520–540 nm and an emission wavelength of 590–620 nm with a gain of 1,900 using a CLARIOstar plate reader (BMG LABTECH) in 384-well F-bottom FLUOTRAC™ nonbinding microplates (Greiner Bio-One). Core TFIIH was assembled with all subunits present in equimolar amounts with a final concentration of 500 nM and equilibrated on ice for 45 min. Assays were performed at 30 °C in 20 mM HEPES pH 7.5, 4.5% (v/v) glycerol, 135 mM KCl, 9 mM $MgCl_2$, 2 mM phosphoenolpyruvate, 0.7 mM TCEP and 1.62 U PK. Initial velocities of the kinetic data curves were fitted with the MARS software package (BMG LABTECH) and represent the averages of at least three technical replicates and one biological replicate. Mean values were plotted with their associated s.d. The mean and s.d. were determined using GraphPad Prism software.

### Differential scanning fluorimetry

Correct folding of the XPD variants was tested by thermal shift assays using SYPRO Orange (Invitrogen) and a qPCR machine (Stratagene mx3005p). The final reaction mix of 25 µl comprised 2.5 µM XPD, 0.1% SYPRO Orange, 20 mM HEPES pH 7.5, 200 mM NaCl, 5 mM $MgCl_2$ and 1 mM TCEP. Unfolding was observed as an increase in fluorescence that was detected at an excitation wavelength of 492 nm and an emission wavelength of 610 nm. Data were plotted using GraphPad Prism and represent the average of at least three different measurements.

### Reporting summary

Further information on research design is available in the Nature Portfolio Reporting Summary linked to this article.

## Data availability

Cryo-EM data and coordinates were deposited to the Electron Microscopy Data Bank (EMDB) and PDB, respectively. The class 1 data and XPD–DNA complex model are available under accession codes EMD-19109 and PDB 8REV. The class 2 data are available under accession code EMD-19109. Other research data will be made available upon request. Source data are provided with this paper.

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

## Acknowledgements

This work used the cryo-EM beamline CM01 at the ESRF and the cryo-EM facility of the Julius-Maximilians-Universität Würzburg funded by the German Research Foundation (359471283, 456578072 and 525040890). This research was supported by the German Research Foundation (KI 562 11-1, CK), (HO4436/4-1, CH) and the German Cancer Aid (70114277, CK). The funders had no role in study design, data collection and analysis, decision to publish or preparation of the manuscript.

## Author contributions

T.H., M.K., T.S., E.G., and S.M. carried out the biochemical studies. E.G. purified all proteins for this study. F.S. and E.G. generated XPD variants. H.N. produced the cross-linked DNA substrate. H.N. and C.H. supervised cross-linked DNA substrate production. T.H. performed sample and cryo-EM grid preparation. J.K. carried out data analysis and cryo-EM data processing. J.K. and C.K. conceptualized the research project. J.K. and C.K. supervised experimental design and data interpretation. J.K. and C.K. led the manuscript writing process. All authors were involved in writing the paper and adhered to the 'inclusion and ethics' regulation.

## Funding

## Competing interests

The authors declare no competing interests.

## Additional information

**Extended data** is available for this paper at https://doi.org/10.1038/s41594-024-01323-5.

**Correspondence and requests for materials** should be addressed to Jochen Kuper or Caroline Kisker.

a

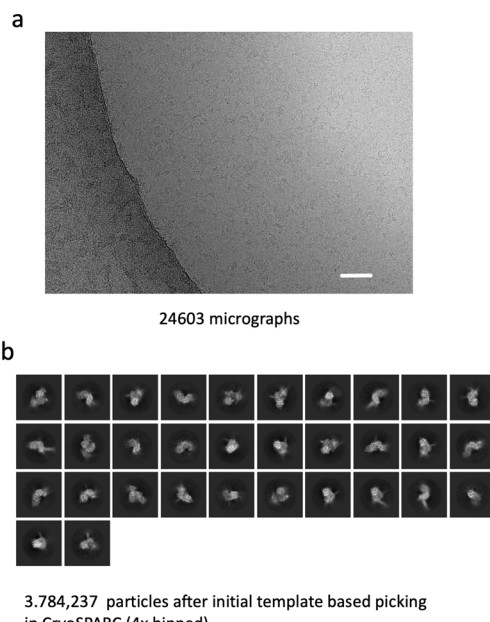

24603 micrographs

b

3.784,237 particles after initial template based picking
in CryoSPARC (4x binned)

c

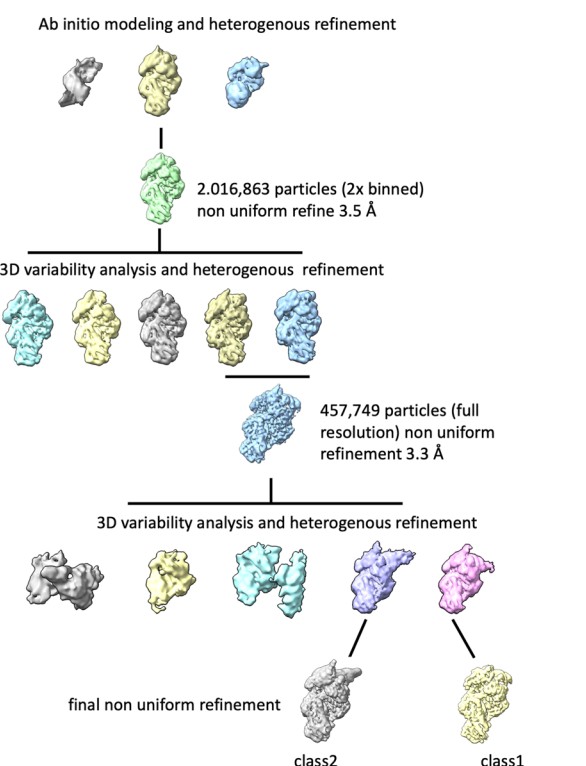

Ab initio modeling and heterogenous refinement

2.016,863 particles (2x binned)
non uniform refine 3.5 Å

3D variability analysis and heterogenous refinement

457,749 particles (full
resolution) non uniform
refinement 3.3 Å

3D variability analysis and heterogenous refinement

final non uniform refinement

class2
121,289 particles 3.4 Å

class1
237, 064 particles 3.1 Å

**Extended Data Fig. 1 | Workflow of cryo EM data processing. a**) Representative micrograph of the XPD complex/DNA sample revealing the single particles to be evenly distributed, n = 24603. The white scalebar represents 50 nm. **b**) Reference free 2D class averages obtained with CryoSPARC. The selected classes have been obtained from the initial round of template based particle picking and represent 3.784,237 particles. **c**) Schematic workflow of data processing in CryoSPARC. All employed steps are indicated.

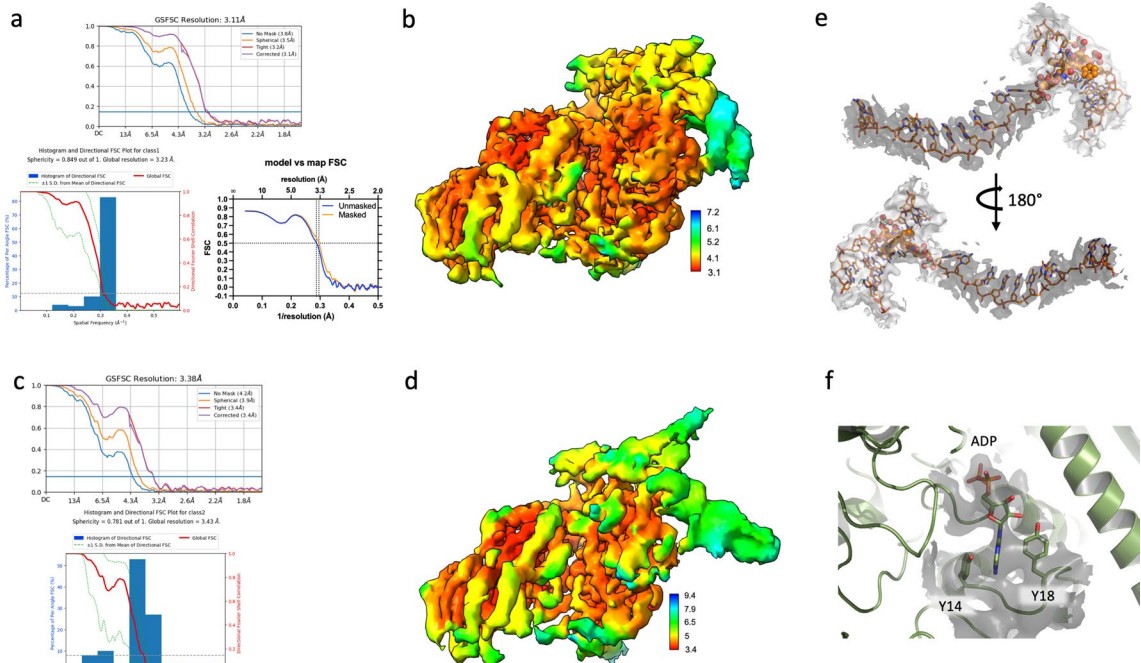

**Extended Data Fig. 2 | Map quality of the resulting cryo EM maps.**
**a**, **c**) GSFSC (for details see methods section) and 3DFSC correlation plots for class 1 (a) and class 2 (c) indicating the resolution limits and anisotropy of the data, respectively. **a**) Contains in addition a model vs map FSC analysis generated with the PHENIX package[41]. **b**, **d**) Local resolution maps of class 1 (b) and class 2 (d) cryo EM maps. **e**) Density of the DNA substrate. Gray density represents the map plotted with 10 sigma, light gray density represents lower resolution data of the same map plotted at 5 sigma. The map carving radius around objects was set to 5 Å. The DNA is shown in cartoon mode. **f**) Density for ADP plotted at 10 sigma. ADP is shown in stick mode, whereas the protein is drawn as cartoon.

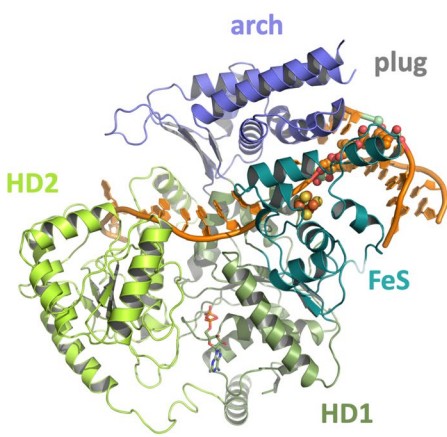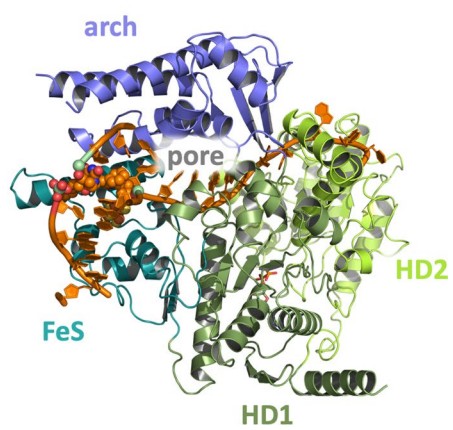

**Extended Data Fig. 3 | Overview of XPD architecture.** The XPD crosslink DNA structure is shown in cartoon representation. Structural elements are highlighted and color coded. Left panel: Front view of XPD. Right panel: left panel rotated 180° around the y-axis. Abbreviations are as follows, arch= XPD Arch domain, HD1 and HD2= helicase motor domains 1 and 2, FeS= XPD iron sulfur cluster domain.

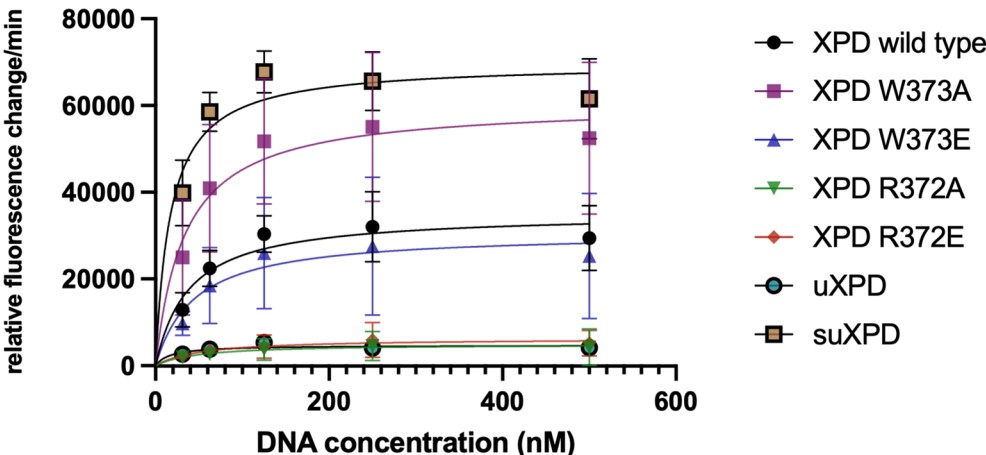

**Extended Data Fig. 4 | Helicase activity of XPD variants.** DNA dependent XPD/p44 helicase activity. DNA was used in a concentration range from 500- 31.25 nM with 1:1 dilutions. Curves were fitted with GraphPad Prism and represent the averages of at least three technical replicates and one biological replicate. Mean values are plotted with their associated SD N values for samples: XPD wild type n = 16, XPD W373A n = 10, XPD W373E n = 15, XPD R372A n = 10, XPD R372E n = 6, uXPD n = 6, suXPD n = 6.

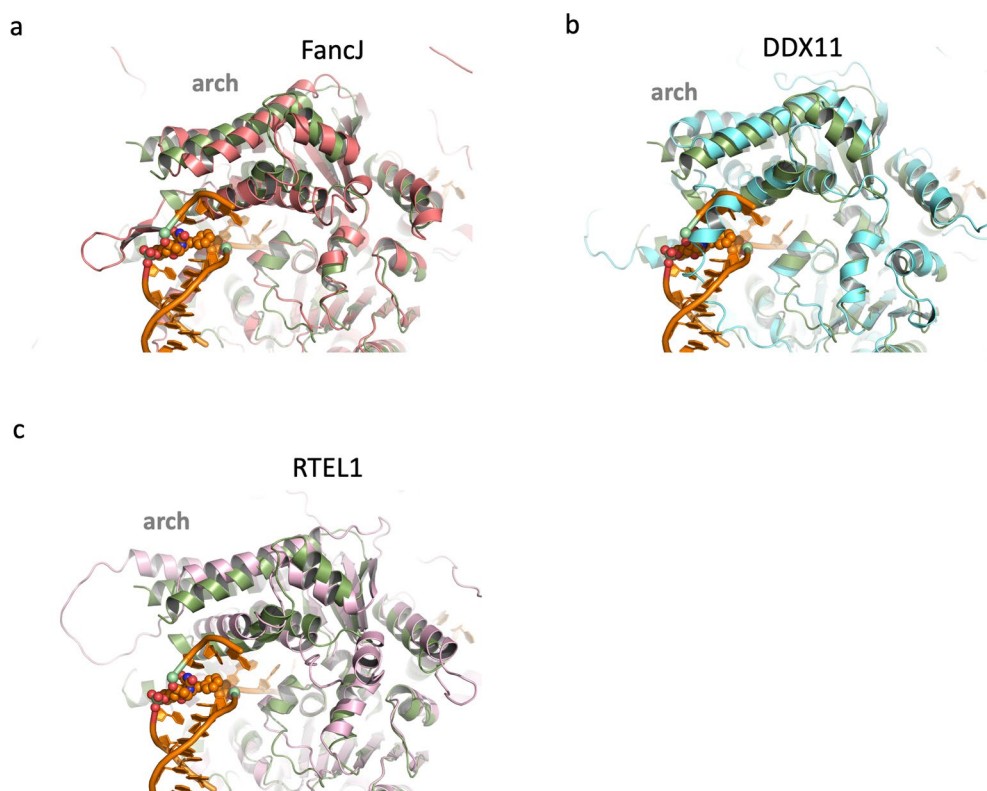

**Extended Data Fig. 5 | FeS containing helicases.** Superposition of the structure obtained in this work with AlphaFold models of **a**) FANCJ (colored in salmon), **b**) DDX11 (colored in cyan), and **c**) RTEL1 (colored in pink). All models are shown as cartoon. All proteins contain an Arch domain suggesting a conserved mode to unwind DNA.

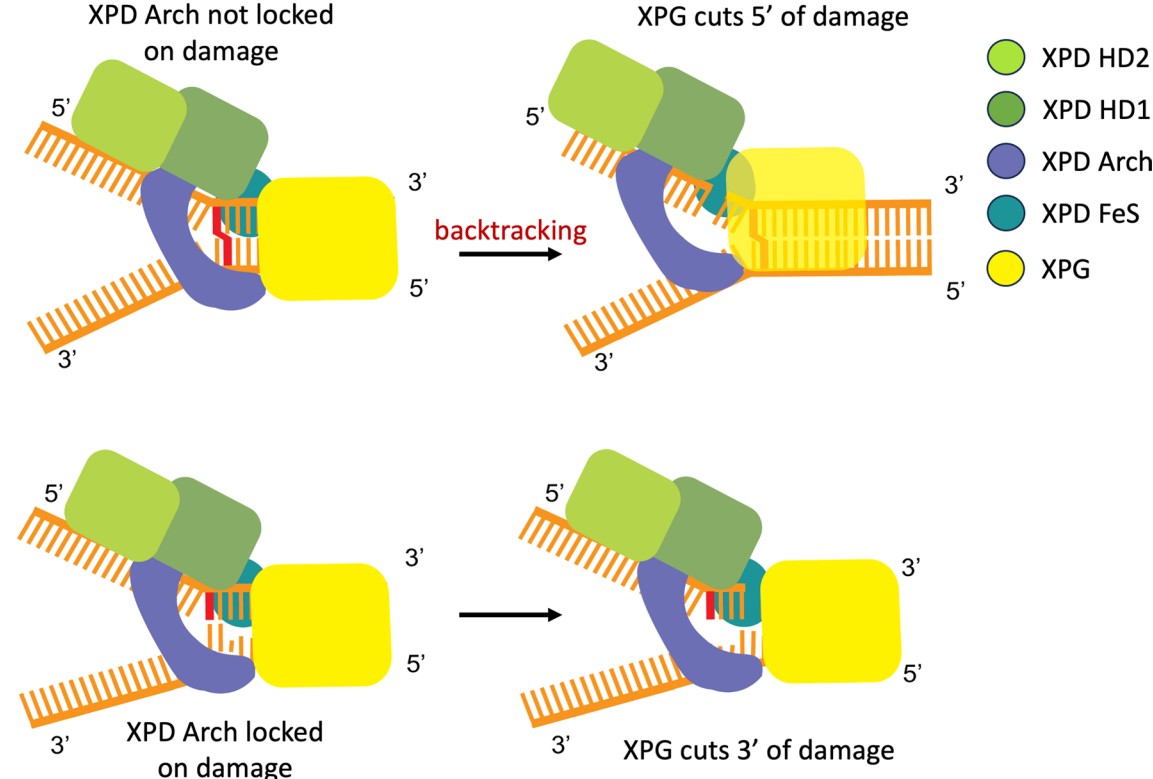

**Extended Data Fig. 6 | Hypothetical lesion recognition model for canonical and non canonical lesions.** The figure shows possible lesion recognition strategies for non canonical and canonical NER lesions. Upper panel: When XPD encounters a non canonical crosslink no lesion locking occurs and XPD backtracking can take place. Backtracking pulls XPG across the lesion enabling cutting 5′ to the lesion. Lower panel: canonical lesion encounter enables XPD lesion locking and XPG cutting 3′ to the damage as observed in regular NER.

**Extended Data Table 1 | Biochemical data**

| Protein | DNA binding kD ($R^2$ of fit) | ATPase µM*min$^{-1}$ ± SD | Helicase ΔF*min$^{-1}$ *1000 ± SD | Melting point °C ($R^2$ of fit) | Translocase ΔF*min$^{-1}$*1000 |
|---|---|---|---|---|---|
| XPD wild type | 30 (0.97) | 16 ±4 | 32 ±8 | 56 (0.69) | 2.1 ±0.9 |
| R372A | 54 (0.84) | 12 ±8 | 5 ±3 | 49 (0.76) | n.d. |
| R372E | 36 (0.88) | 10 ±5 | 6±4 | 54 (0.78) | n.d. |
| W373A | 27 (0.91) | 21 ±8 | 55 ±17 | 51 (0.72) | n.d. |
| W373E | 45 (0.84) | 12 ±6 | 28 ±16 | 50 (0.71) | n.d. |
| uXPD | 30 (0.98) | 12 ± 7 | 5 ±2 | 56 (0.99) | 1.3 ±0.6 |
| suXPD | 37 (0.99) | 11 ±4 | 66 ±7 | 54 (0.88) | 6.0 ±2.7 |

This table shows the values for the biochemical data presented in Figs. 3 and 5b.

**Extended Data Table 2 | XPG dependent XPD helicase activity**

| Protein | Helicase ΔF*min$^{-1}$ *1000 ± SD |
|---|---|
| XPD wild type | 24 ±3 |
| XPD wild type + XPG D924A | 37 ±10 |
| suXPD | 66±4 |
| suXPD + XPG D924A | 21 ±8 |
| uXPD | 5 ±2 |
| uXPD + XPG D924A | 15 ±1 |

This table shows the values for the biochemical data in Fig. 5c.

# Reporting Summary

## Statistics

For all statistical analyses, confirm that the following items are present in the figure legend, table legend, main text, or Methods section.

| n/a | Confirmed | |
|---|---|---|
| ☐ | ☒ | The exact sample size (*n*) for each experimental group/condition, given as a discrete number and unit of measurement |
| ☐ | ☒ | A statement on whether measurements were taken from distinct samples or whether the same sample was measured repeatedly |
| ☐ | ☒ | The statistical test(s) used AND whether they are one- or two-sided *Only common tests should be described solely by name; describe more complex techniques in the Methods section.* |
| ☒ | ☐ | A description of all covariates tested |
| ☒ | ☐ | A description of any assumptions or corrections, such as tests of normality and adjustment for multiple comparisons |
| ☐ | ☒ | A full description of the statistical parameters including central tendency (e.g. means) or other basic estimates (e.g. regression coefficient) AND variation (e.g. standard deviation) or associated estimates of uncertainty (e.g. confidence intervals) |
| ☐ | ☒ | For null hypothesis testing, the test statistic (e.g. *F*, *t*, *r*) with confidence intervals, effect sizes, degrees of freedom and *P* value noted *Give P values as exact values whenever suitable.* |
| ☒ | ☐ | For Bayesian analysis, information on the choice of priors and Markov chain Monte Carlo settings |
| ☒ | ☐ | For hierarchical and complex designs, identification of the appropriate level for tests and full reporting of outcomes |
| ☒ | ☐ | Estimates of effect sizes (e.g. Cohen's *d*, Pearson's *r*), indicating how they were calculated |

*Our web collection on statistics for biologists contains articles on many of the points above.*

## Software and code

Policy information about availability of computer code

| Data collection | EPU software (ThermoFisher) was used for data collection |
|---|---|
| Data analysis | We have used GraphPadPrism (10.2.1) for data analysis and statistics of biochemical data. For single particle data processing we have used the cryoSPARC suite (4.2.x). For refinement and model building COOT (0.9.8.4), refmac5 (5.8.0352), and the PHENIX suite (1.21rc1-5127) have been used. |

For manuscripts utilizing custom algorithms or software that are central to the research but not yet described in published literature, software must be made available to editors and reviewers. We strongly encourage code deposition in a community repository (e.g. GitHub). See the Nature Portfolio guidelines for submitting code & software for further information.

## Data

Policy information about availability of data

All manuscripts must include a data availability statement. This statement should provide the following information, where applicable:
- Accession codes, unique identifiers, or web links for publicly available datasets
- A description of any restrictions on data availability
- For clinical datasets or third party data, please ensure that the statement adheres to our policy

Cryo EM data and coordinates have been deposited with EMDB and pdb, respectively. Class 1 and the XPD/DNA complex model are available under the accession

codes EM-19109 and 8rev. Class 2 data are available under EM-19109. Other research data will be made available upon request since the source data for the manuscript have been supplied with the current manuscript. We have used following data from the pdb: 6ro4, 6nmi, and 4q0w.

## Research involving human participants, their data, or biological material

Policy information about studies with [human participants or human data](). See also policy information about [sex, gender (identity/presentation), and sexual orientation]() and [race, ethnicity and racism]().

| | |
|---|---|
| Reporting on sex and gender | not applicable |
| Reporting on race, ethnicity, or other socially relevant groupings | not applicable |
| Population characteristics | not applicable |
| Recruitment | not applicable |
| Ethics oversight | not applicable |

Note that full information on the approval of the study protocol must also be provided in the manuscript.

## Field-specific reporting

Please select the one below that is the best fit for your research. If you are not sure, read the appropriate sections before making your selection.

☒ Life sciences ☐ Behavioural & social sciences ☐ Ecological, evolutionary & environmental sciences

For a reference copy of the document with all sections, see nature.com/documents/nr-reporting-summary-flat.pdf

## Life sciences study design

All studies must disclose on these points even when the disclosure is negative.

| | |
|---|---|
| Sample size | At least three technical and one biological replicate if not mentioned otherwise. The sample size was determined by intensive pre characterization of the assay systems under various conditions leading to reliable and reproducible results. |
| Data exclusions | no data was excluded |
| Replication | all replicates were successful |
| Randomization | The assay systems were developed by testing a broad range of parameters to ensure reliable results and have been published in many peer reviewed articles |
| Blinding | Assays were performed independently by different persons yielding comparable results based upon the existing protocols |

## Reporting for specific materials, systems and methods

We require information from authors about some types of materials, experimental systems and methods used in many studies. Here, indicate whether each material, system or method listed is relevant to your study. If you are not sure if a list item applies to your research, read the appropriate section before selecting a response.

### Materials & experimental systems

| n/a | Involved in the study |
|---|---|
| ☒ | Antibodies |
| ☐ | ☒ Eukaryotic cell lines |
| ☒ | Palaeontology and archaeology |
| ☒ | Animals and other organisms |
| ☒ | Clinical data |
| ☒ | Dual use research of concern |
| ☒ | Plants |

### Methods

| n/a | Involved in the study |
|---|---|
| ☒ | ChIP-seq |
| ☒ | Flow cytometry |
| ☒ | MRI-based neuroimaging |

# Eukaryotic cell lines

Policy information about cell lines and Sex and Gender in Research

| | |
|---|---|
| Cell line source(s) | Insect cell lines were obtained from ThermoFisher |
| Authentication | Cell lines were authenticated by vendor |
| Mycoplasma contamination | Cell lines were not tested, since they were only used for protein production. Proteins were then purified to homogeneity and used for assays. |
| Commonly misidentified lines<br>(See ICLAC register) | not applicable |

