## [Peer Review File · Nature Structural & Molecular Biology]

Peer Review Information

Manuscript Title: XPD stalled on crosslinked DNA provides insight into damage verification

Corresponding author name(s): Caroline Kisker, Jochen Kuper

Editorial Notes:

Transferred manuscripts (no peer review at Nature Structural & Molecular Biology) This manuscript has been previously reviewed at another journal that is not operating a transparent peer review scheme. The manuscript was considered suitable for publication without further review at Nature Structural & Molecular Biology.

Reviewer Comments & Decisions:

Decision Letter, initial version:

Message: Our ref: NSMB-A48928-T

12th Mar 2024

Dear Dr. Kisker,

Thank you for submitting your revised manuscript "XPD stalled on crosslinked DNA provides insight into damage verification" (NSMB-A48928-T). We have read and assessed the response to the remaining concerns of the original referee, with the other two having already assessed that the manuscript is ready for publication, and we editorially deem that all raised concerns have been addressed. Therefore we are happy to accept the manuscript in principle in Nature Structural & Molecular Biology, pending minor revisions to comply with our editorial and formatting guidelines.

We are now performing detailed checks on your paper and will send you a checklist detailing our editorial and formatting requirements in about two weeks. Please do not upload the final materials and make any revisions until you receive this additional information from us.

To facilitate our work at this stage, it is important that we have a copy of the main text as a word file. If you could please send along a word version of this file as soon as possible, we would greatly appreciate it; please make sure to copy the NSMB account (cc'ed above).

Sincerely,

Dimitris Typas
Associate Editor
Nature Structural & Molecular Biology
ORCID: 0000-0002-8737-1319

Final Decision Letter:

Message: 24th Apr 2024

Dear Dr. Kisker,

We are now happy to accept your revised paper "XPD stalled on crosslinked DNA provides insight into damage verification" for publication as an Article in Nature Structural & Molecular Biology.

Your paper will be published online soon after we receive proof corrections and will appear in print in the next available issue. You can find out your date of online publication by contacting the production team shortly after sending your proof corrections.

Please note that *Nature Structural & Molecular Biology* is a Transformative Journal (TJ). Authors may publish their research with us through the traditional subscription access route or make their paper immediately open access through payment of an article-processing charge (APC). Authors will not be required to make a final decision about

access to their article until it has been accepted. Find out more about Transformative Journals

Sincerely,

Dimitris Typas
Associate Editor
Nature Structural & Molecular Biology
ORCID: 0000-0002-8737-1319